# Structural, Optical, and Magnetic Studies of the Metallic Lead Effect on MnO_2_-Pb-PbO_2_ Vitroceramics

**DOI:** 10.3390/ma15228061

**Published:** 2022-11-15

**Authors:** Simona Rada, Mihaela Unguresan, Mioara Zagrai, Adriana Popa

**Affiliations:** 1Physics and Chemistry Department, Technical University of Cluj-Napoca, 400020 Cluj-Napoca, Romania; 2National Institute for Research and Development of Isotopic and Molecular Technologies, 400293 Cluj-Napoca, Romania

**Keywords:** MnO_2_-Pb-PbO_2_, XRD, IR, UV–Vis, EPR

## Abstract

MnO_2_-lead materials have attracted attention in their applications as electrodes. This work reports a detailed spectroscopic study of the compositional variation of MnO_2_-xLead vitroceramic materials with varied Pb contents. The concentration variation of lead and manganese ions issystematically characterized throughthe analysis of X-ray diffraction (XRD), Fourier transform infrared (FTIR), ultraviolet–visible (UV–Vis), and electron paramagnetic resonance (EPR) spectroscopy.The MnO_2_-xLead samples consist of a vitroceramic structure with Pb, PbO, PbO_2_,and Mn_3_O_4_ crystalline phases. The introduction of higher Pb content in the host vitroceramic reveals the [PbO_6_]→[PbO_n_] conversion, where *n* = 3, 4, and the formation of distorted [MnO_6_] octahedral units. The UV–Vis data of the samples possess the intense bands between 300 and 500 nm, which are due to the presence of divalent lead ions (320 nm) and divalent and trivalent manganese ions (420 and 490 nm, respectively) in the structure of glass ceramics. The EPR data show resonance lines located around g ~ 8 and 4.3, and a sextet hyperfine structure at g ~ 2, which isascribed to the Mn^+3^ and Mn^+2^ ions.

## 1. Introduction

Lead-containing glass and vitroceramics can be produced viathe melt-quenching method at a low temperature [1,2,3]. Nowadays, lead glass and vitroceramicscontainingdifferent transition metal ions can be used in optoelectronics andacousto-optics for the production of devices and electrodes in batteries [4,5,6,7,8].

The manganese ions may enhance the chemical resistance of the lead network due to its unique chemistry. Different manganese ions, such as Mn^+2^ and Mn^+3^ ions, are known as paramagnetic ions. In glass and vitroceramics, the amount of these species depends on the mobility of the cations, field strength, condition of melting, concentrationof manganese in the host matrix, glassy modifiers, and formers [9].

Manganese oxide materials, namely MnO_2_ and Mn_3_O_4_, are considered some of the most promising electrode materials because they are inexpensive, eco-friendly, and have a highly specific capacitance and cycling stability [10].

Oxide glass and vitroceramics doped with transition metal ions, such as MnO_2_, are technologically important because theycontain semiconduction and photo-conduction properties, switchingbehavior, and good optical absorption [11]. The tetrahedral or octahedral geometries of the manganese ions incorporated into the host network by varying the composition of glass or vitroceramics can be performed in the EPR and UV–Vis spectra.

In order to better understand manganese-lead glass and vitroceramics from an application point of view, the study of structure is necessary. The MnO_2_-Pb-PbO_2_ vitroceramics can be used as electrodes for alead acid battery [8]. The present paper evidences structural information on the MnO_2_-xLead vitroceramic materials with x = 0–100 mol% Pb. The molar percentage of manganese dioxide was constantly 15 mol% MnO_2_. The effect of the substitution of lead dioxide by metallic lead on the host matrix was investigated through the analysis of XRD, IR, and UV–Vis spectroscopy. In this context, the variation of manganese ions’valence states as a function of Pb concentration was also studied using EPR spectroscopy. The role of the lead level on the structural aspects of MnO_2_-xLead vitroceramics materials using a systematic study on spectroscopic characteristics wasalso investigated.

## 2. Experimental Procedure

The MnO_2_-xLead samples with x = 0–100 mol% Pb were prepared from MnO_2_, PbO_2_,and metallic Pb powders. The fine mixtures of substances were melted in a sintered alumina crucible using an electric furnace set to 850 °C for 10 min. The melt was rapidly cooled on a steel plate.

The flow diagram of the synthesis and the Fujifilm images of the obtained samples are shown in Figure 1. The samples have a metallic appearance and are black in color. The composition of the MnO_2_-xLead prepared samples is listed in Table 1.

X-ray diffractograms were obtained at room temperature witha Shimadzu XRD-6000 diffractometer using the fine powder sample and an increment of 0.02 ^o^·s^−1^.

IR absorption spectra were recorded using the JASCO 6200 FTIR spectrometer. The FTIR measurements were carried out at a resolution of 4 cm^−1^.

UV–Visible spectra of the samples were recorded using a Perkin-Elmer Lambda 45 UV/VIS spectrometer with an integrated sphere and a resolution of 2 nm. The sample-to-KBr ratio was 1:150.

Electron paramagnetic resonance (EPR) spectroscopy measurements were performed using a Bruker ELEXSYS 500 spectrometer in X-band. The samples were introduced in the glass tube using equal amounts of samples.

## 3. Results and Discussion

### 3.1. Structural Investigation by X-ray Diffraction (XRD)

X-ray patterns of the MnO_2_-xLead samples are shown in Figure 2. The XRD data confirm the vitroceramic structure of all prepared samples. The diffraction peaks correspond to the Pb crystalline phase (pdf. no. 00-004-0686) as the main phase and small amounts ofPbO_2_ (00-052-0752), PbO (01-072-0093), and Mn_3_O_4_ (pdf. no. 00-018-803) crystalline phases.

### 3.2. Structural Investigations by IR Spectroscopy

The MnO_2_-xLead vitroceramic materials, where x = 0–100 mol% Pb, were investigated by IR spectroscopy in order to obtain information concerning the compositional evolution of the structural units with the addition of the Pb content in the vitroceramic matrix. The obtained IR spectra are shown in Figure 3. Wavenumbers and the assignment of the IR bands characteristic of different structural units of the vitroceramic matrix are listed in Table 2.

The IR band situated at ~470 cm^−1^ came from two contributions, namely, the deformation vibrations of the Pb-O-Pb and O-Pb-O angles in the [PbO_4_] units [12] superimposed with the bending vibrations of the O-Mn-O angles from [MnO_6_] octahedral units [13]. The intensity of this IR band increased up to a maximum value for the sample with 50 mol% Pb.

The IR band centered at 880 cm^−1^ corresponded to the stretching vibrations of the Pb-O bonds in the [PbO_6_] units.

The IR band located at 520 cm^−1^ corresponded to the stretching vibrations of the Mn-O bonds from distorted octahedral units. After introducing the dopant level up to x ≤ 90 mol% Pb, this IR band became well-formed, and its intensity was enriched slightly.

The IR band located at 580 cm^−1^ corresponded to the deformation vibrations of the Mn-O-Mn angles. This IR band appeared well-formed for samples with x ≤ 40 mol% Pb. The PbO_2_-MnO_2_ consisted of [PbO_n_] structural units with *n* = 3, 4, and 6, as well as distorted octahedral [MnO_6_] units between them, connected by Pb-O-Pb and Mn-O-Mn bridges. Throughthe addition oflower Pb contents up to x ≤ 40 mol%, the amount of these structural units increased due to the excess of non-bonding oxygen. By increasing the dopant level above x ≥ 50 mol% Pb, the Mn-O-Mn linkages were broken and the affinity of lead atoms towards unbounded oxygen atoms from distorted [MnO_6_] octahedral units produced an increase in the fractions of [PbO_n_] structural units with *n* = 3, 4, and 6 reaching maximum values for the sample with x = 70 and 80 mol% Pb.

The second region of intense IR bands, situated between 750 and 1100 cm^−1^, corresponded to the stretching vibrations of the Pb-O bond in the [PbO_n_] units (*n* = 3 and 4). Through the gradual addition of metallic lead in the host matrix up to x ≤ 40 mol%, the intensity of the bands located in this region increased and showeda tendency to move towards higher wave numbers. The excess oxygen can be accommodated in the host matrix by the formation of [PbO_n_] structural units with *n* = 3, 4, and 6. For samples with x ≥ 50 mol% Pb, a tendency of displacement of the IR bands towards higher wave numbers was evidenced, indicating the increase of [PbO_n_] structural units with *n* = 3 and 4.

The IR band from 875 cm^−1^ was assigned to the [PbO_6_] structural units. The intensity of this IR bandincreased up to x ≤ 70 mol% Pb, andthen decreased and reached a maximum value for the sample with x = 70 mol% Pb.

For the sample with 100 mol% Pb containing the 0.15MnO_2_·0.85Pb composition, the formation of a glass–ceramic structure was identified due to the ability of lead atoms to form [PbO_n_] units (some structural units [PbO_6_]-band IR from 875 cm^−1^ decreased in intensity) and manganese to form distorted [MnO_6_] structural units interspersed between Pb-O-Pb linkages.

For the validation of the results, a deconvolution procedure of IR spectra was applied using a Gaussian-type function with fit multi-peaks of the Origin software. The deconvoluted IR spectrum of the sample with 30 mol% Pbis shown in the Figure 3c. The parameters of deconvolution, such as location of the center, C and integral intensity, A of detected IR peaks of deconvolution, are summarized in Table 3. Information concerning the compositional evolution of [MnO_6_], [PbO_6_], and [PbO_n_] structural units, as well as Pb-O-Pb and Mn-O-Mn bridges with increasing Pb content, can be obtained from their integral intensity. Our results show that the integral intensities of these structural units depend on the Pb content.

The fractions of [PbO_6_] structural units give the maximum values for the samples with x = 20, 50, and 90 mol% Pb. The integral intensity of the [PbO_3_] structural units decreased abruptly for the sample with x = 50 mol% Pb, and after that, its area increased up to 100 mol% Pb. The areas of the Pb-O-Pb and Mn-O-Mn bridges were enriched for the vitroceramics with x = 60 and 80 mol% Pb, respectively. A trend of formation of the Mn-O-Mn bridges can be evidenced up to 60 mol% Pb when the enrichment of the integral intensity of [MnO_6_] structural units was reached by adding a Pb level up to 80 and 90 mol%.

The IR data indicate the following structural modifications through the introduction of varied dopant levels: (i) For low Pb contents up to x ≤ 40 mol%, the host vitroceramic adapts with the excess lead by involving it in the formation of [PbO_n_] polygons (*n* = 3 and 4) and distorted [MnO_6_] octahedral units. The polymerization degree of the lead–manganese vitroceramic increased and the number of Mn-O-Mn fractions attained a maximum value; (ii) By increasing the lead content from 50 ≤ x ≤ 70 mol% Pb, the Mn-O-Mn bonds in the host vitroceramicwere broken and the number of [PbO_6_] structural units increased; (iii) At high dopant concentrations 70 < x ≤ 90 mol% Pb, the inadequacy of [PbO_6_] octahedral units with the excess oxygen will produce their conversion into [PbO_n_] structural units (*n* = 3 and 4). (iv) The Pb-MnO_2_ matrix consists mainly of [PbO_n_] structural units and some distorted [MnO_6_] octahedral units.

The structural effect of Pb in MnO_2_-Pb-PbO_2_ vitroceramics is distinguished by the role as network former and modifier. At smaller Pbcontents, the lead atoms prefer [PbO_4_] tetrahedral units and distorted [PbO_6_] structural units. This is accompanied by the simultaneous conversion of [MnO_4_] into [MnO_6_] structural units. At higher Pb levels, glass–ceramics are characterized by the presence of a number of [PbO_n_] polyhedrons (*n* = 3 and 4), which act as a network modifier and remain in the structure as Pb^+2^ ions.

### 3.3. Structural Investigation by UV–Vis Spectroscopy

The UV–Vis spectra of the MnO_2_-xLead samples are shown in Figure 4. The first region of UV–Vis bands located between 200 and 300 nm is due to n-π^*^ transitions of the Pb = O bond in the [PbO_3_] structural units. The second region of UV–Vis bands located between 300 and 500 nm is due to the presence of Pb^+2^ (320 nm), Mn^+2^ (420 nm) [13,14], and Mn^+3^ (490 nm) ions [15] in the structure of glass ceramics. The band situated at about 420 nm is attributed to the transition of ^6^A_1_(S) to ^4^A_1_ of Mn^+2^ ions [16].

By increasing the Pb content over 70 mol%, the intensity of UV–Vis bands located in the range between 300 and 440 nm was enriched. This indicates an increase in the fractions of Pb^+2^ and Mn^+2^ ions by increasing the dopant level.

### 3.4. Optical Band Gap Energy

The variation of (αhν)^1/2^ and (αhν)^2^ as functions of photon energy, *hν*, and the dependence of the x mol% Pb on the optical band gap energy, Eg, for the MnO_2_-xLead samples are plotted in Figure 5 and Figure 6. The value of Eg was determined by extrapolating the linear domain of the (αhν)^1/2^ and (αhν)^2^ graphs as a function of hν at αhν→ 0 [17]. The E_g_ values are situated between 2.47 and 2.65 eV for indirect transitions (with *n* = 2) and between 2.2–2.49 eV for direct transitions (with *n* = 1/2). This indicates a semiconductor behavior (gap energy, Eg < 3 eV) for all samples.

### 3.5. Structural Investigation by EPR Spectroscopy

The local geometry of the two manganese paramagnetic ions Mn^+2^ and Mn^+3^ was characterized by EPR spectroscopy. The EPR spectra of the MnO_2_-xLead vitroceramic materials are shown in Figure 7. Our EPR data indicate that the local geometry around the paramagnetic ions depends on the Pb content of the host vitroceramic. The EPR spectra show three resonance lines:two absorption signals centered at g ~ 2 and g ~ 4.3 corresponding to the Mn^+2^ ions [18,19,20], and the resonance line located at g ~ 8 is attributed to Mn^+3^ ions.

The resonance line centered at g ~ 2 was assigned to isolated Mn^+2^ ions located in sites with distorted octahedral geometry, as well as to those involved in dipole and/or superexchange magnetic interactions. Through the addition of metallic Pb up to x ≤ 60 mol% in vitroceramics, this resonance line consisted of a poorly hyperfine structure superposed over a wider resonance line corresponding to the clustered Mn^+2^ ions. For x ≥ 70 mol% Pb, this line became broader and more intense, becoming predominant in the EPR spectrum. This fact highlights the clustered nature of the main fraction of Mn^+2^ ions.

The Mn^+2^ ions at low manganese concentration had a hyperfine structure consistingof six resonance lines, which result from the dipole–dipole interaction between the magnetic moment of the ^55^Mn nucleus and the electronic moment of the paramagnetic Mn^+2^ ion [16].

For the samples with x ≤ 60 mol%, the gradual disappearance of the hyperfine structure in the vitroceramics by doping with Pb content suggested the improvement of dipole–dipole interactions. The addition of higher Pb concentrations (x ≥ 70 mol% Pb) indicates an increase in magnetic exchange interaction.

The resonance signal located at g ~ 4.3 corresponded to a rhombic distorted geometry of isolated Mn^+2^ ions. The intensity of this line decreased slightly with the addition of metallic Pb up to x ≤ 70 mol%, after which it disappeared. For higher dopant concentrationsabove 70 mol% Pb, the Mn^+2^ ions situated in isolated rhombic or distorted octahedral positions became clustered manganese ions. At higher metallic lead contents, x ≥ 80 mol%, the structural disorder around the Mn^+2^ ions increased, and these ions were present only as clustered Mn^+2^ species.

The resonance line located at g ~ 8 was given by the Mn^+3^ ions (at ~800G in the X band) [20]. Its intensity decreasedwith the introduction of Pb levels up to x ≤ 70 mol%, and for higher concentrations, the resonance line was not detected. This structural evolution is explained through the Mn^+3^→Mn^+2^ conversion (according to the UV–Vis data) by increasing the Pb content over x ≥ 70 mol%.

In conclusion, the effect of the Pb content on the host vitroceramic can be presented by two mechanisms: (i) For the vitroceramics with x ≤ 70 mol% Pb, manganese ions are present in two valence states, Mn^+2^ and Mn^+3^. Mn^+3^ ions are obtained during synthesis, upon sudden cooling of the melt. There is a local ordering process of the oxygen atoms around the Mn^+2^ ion; (ii) For the vitroceramics with x ≥ 80 mol% Pb, only clustered Mn^+2^ ions predominated in the studied vitroceramics.

## 4. Conclusions

In this paper, the structural properties of the MnO_2_-xLead vitroceramic materials were investigated in order to find the role of lead content in the host matrix. The structural investigations of the prepared samples were realized by XRD, FTIR, UV–Vis, and EPR spectroscopy. The structure of the vitroceramic materials containingthe 0.15MnO_2_·0.85Pb, 0.15MnO_2_·0.85PbO_2_, and 0.15MnO_2_·0.85[(1−x)PbO_2_·xPb] compositions depends on the lead content.

XRD data indicatedvitroceramic structures for MnO_2_-xLead samples. The analysis of IR data evidenced that the structures are mainly composed of [PbO_n_] and [MnO_n_] structural units. The presence of the lead and manganese ions were evidenced in the UV–Vis spectra.

The EPR data indicated three resonance lines. The resonance lines located at g ~ 2 and 4.3 are due to Mn^+2^ ions, while the last resonance signal corresponds to the Mn^+3^ ions.

## Figures and Tables

**Figure 1 materials-15-08061-f001:**
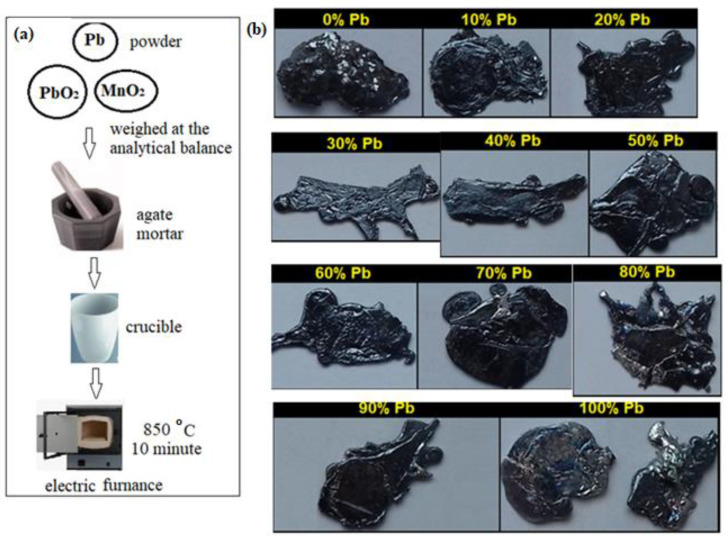
(**a**) Flow diagram of the synthesis; (**b**) Photographic images of prepared MnO_2_-xLead materials. Fujifilm photographic images of prepared MnO_2_-xLead materials.

**Figure 2 materials-15-08061-f002:**
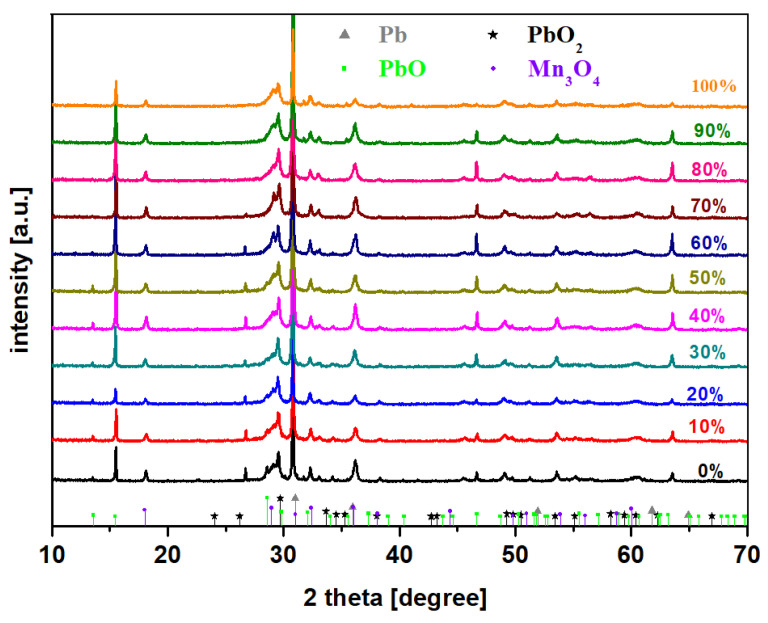
X-ray diffractograms for the prepared MnO_2_-xLead materials, where x = 0–100 mol% Pb.

**Figure 3 materials-15-08061-f003:**
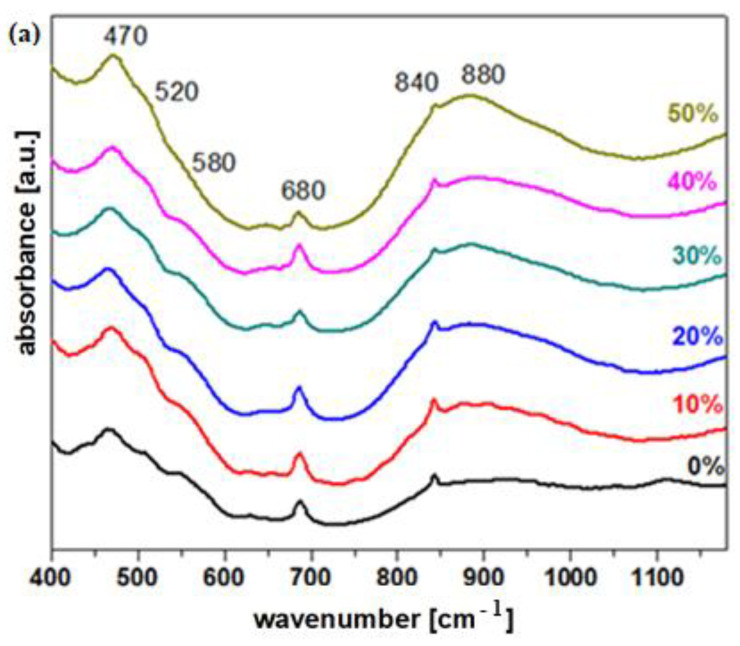
FTIR spectra of the MnO_2_ - xLead vitroceramic materials, where (**a**) x = 0–50 mol% Pb and (**b**) x = 50–100 mol% Pb. (**c**) Deconvoluted IR spectrum for the sample with x = 30 mol% Pb.

**Figure 4 materials-15-08061-f004:**
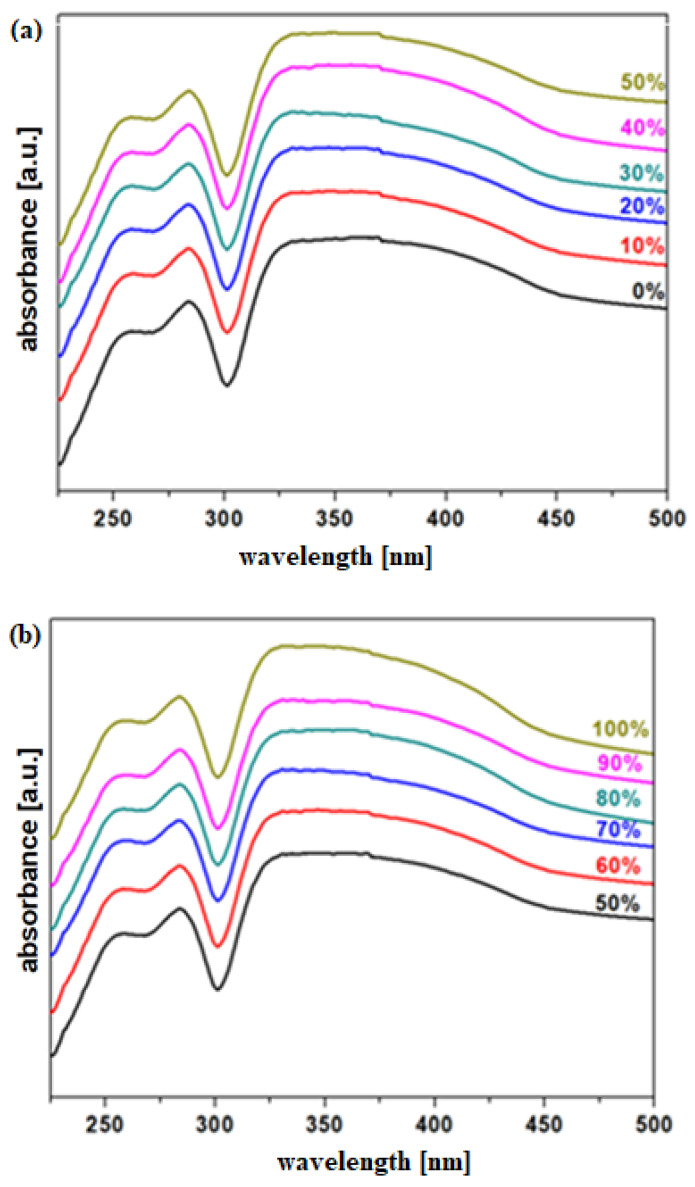
UV–Vis spectra of the MnO_2_-xLead vitroceramic materials (**a**) x = 0–50 mol% Pb (**b**) x = 50–100 mol% Pb.

**Figure 5 materials-15-08061-f005:**
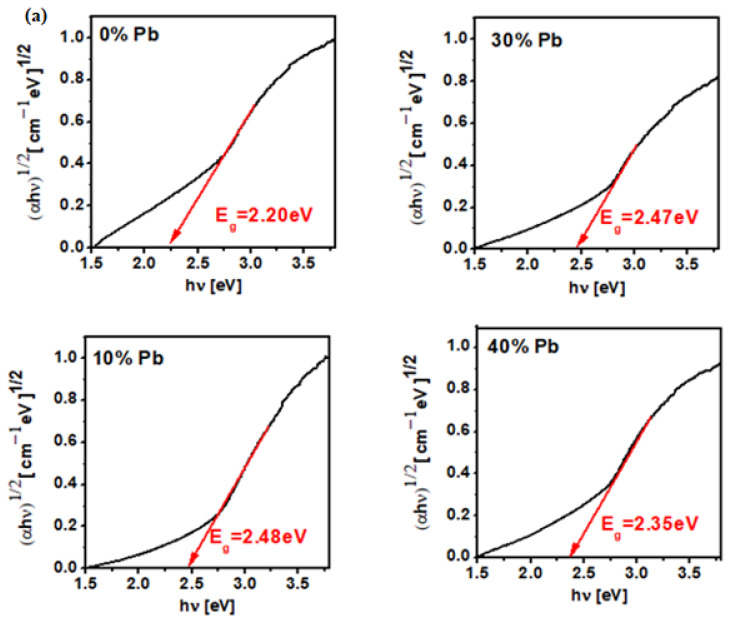
Dependence of (αhν)^1/2^ as function of hν and the dependence of x mol% Pb on gap energy values for the MnO_2_ -xLeadvitroceramic materials (**a**) x = 0–50 mol% Pb (**b**) x = 50–100 mol% Pb.

**Figure 6 materials-15-08061-f006:**
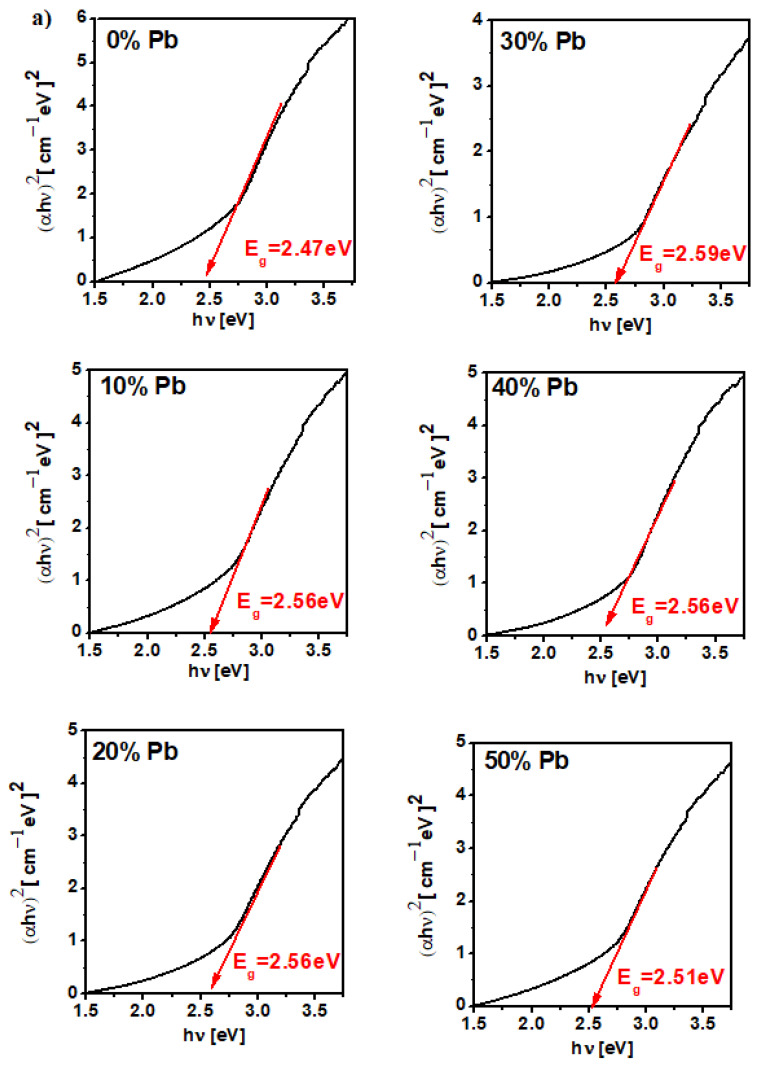
Dependence of (αhν)^2^ as function of hν and the dependence of x mol% Pb on gap energy values for the MnO_2_-xLead vitroceramic materials (**a**) x = 0–50 mol% Pb (**b**) x = 60–100 mol% Pb.

**Figure 7 materials-15-08061-f007:**
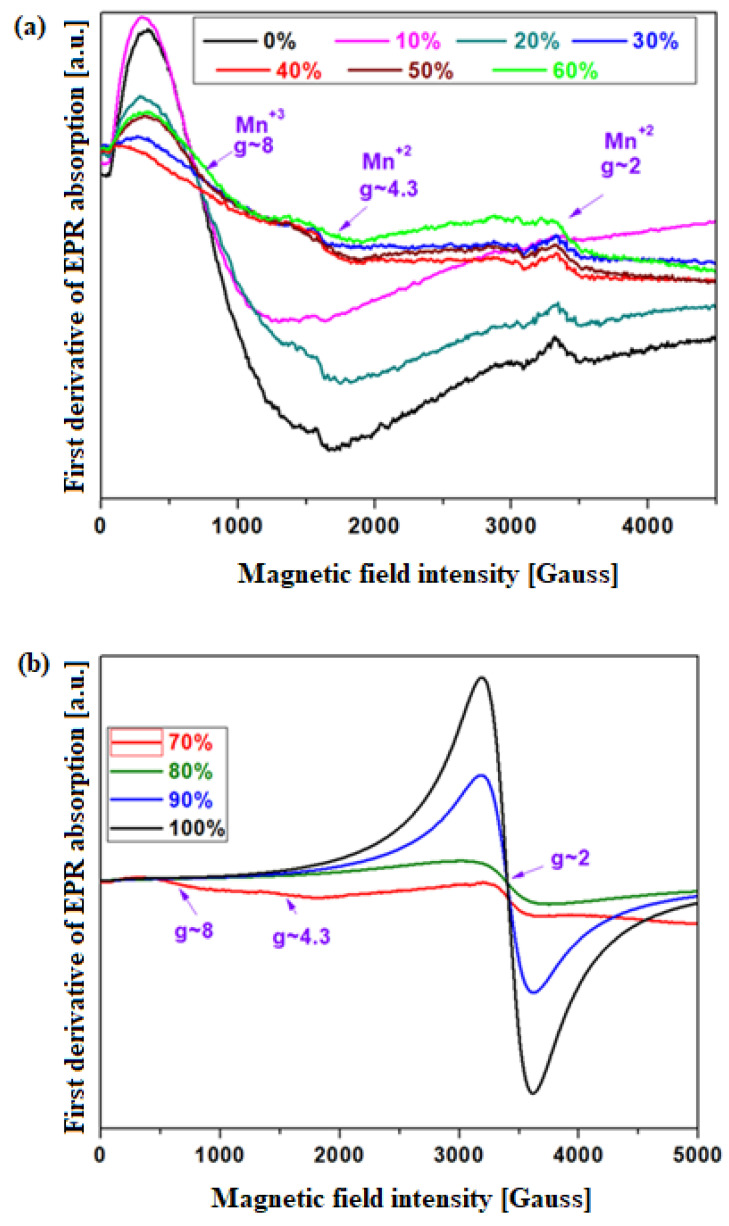
EPR spectra of the MnO_2_-xLead vitroceramic materials. (**a**) x = 0–60 mol% Pb (**b**) x = 70–100 mol% Pb.

**Table 1 materials-15-08061-t001:** Description of prepared MnO_2_-xLead materials.

Notation of MnO_2_-xLead Vitroceramic Materials	Composition of MnO_2_-xLead Vitroceramic Materials
0.15MnO_2_·0.85[(1−x)PbO_2_·xPb]	
x = 0 mol% Pb	0.15MnO_2_·0.85PbO_2_
x = 10 mol% Pb	0.15MnO_2_·0.85[0.9PbO_2_·0.1Pb]
x = 20 mol% Pb	0.15MnO_2_·0.85[0.8PbO_2_·0.2Pb]
x = 30 mol% Pb	0.15MnO_2_·0.85[0.7PbO_2_·0.3Pb]
x = 40 mol% Pb	0.15MnO_2_·0.85[0.6PbO_2_·0.4Pb]
x = 50 mol% Pb	0.15MnO_2_·0.85[0.5PbO_2_·0.5Pb]
x = 60 mol% Pb	0.15MnO_2_·0.85[0.4PbO_2_·0.6Pb]
x = 70 mol% Pb	0.15MnO_2_·0.85[0.3PbO_2_·0.7Pb]
x = 80 mol% Pb	0.15MnO_2_·0.85[0.2PbO_2_·0.8Pb]
x = 90 mol% Pb	0.15MnO_2_·0.85[0.1PbO_2_·0.9Pb]
x = 100 mol% Pb	0.15MnO_2_·0.85Pb

**Table 2 materials-15-08061-t002:** Wave numbers and IR band assignments for the MnO_2_-xLead vitroceramic materials.

Wave Number [cm^−1^]	IR Band Assignment
370–400	Stretching (elongation) vibrations of the Mn^+3^-O bond
470	Deformation (bending) vibrations of Pb-O-Pb and O-Pb-O angles in [PbO_4_] structural units
480	Deformation vibrations of O-Mn-O angles in octahedral [MnO_6_] structural units
520	Specific vibrations of different types of Mn-O bonds in distorted octahedra
580–600	Vibrations specific to Mn-O-Mn type bonds
650–850	Elongation vibrations of the Pb-O bond in [PbO_n_] structural units where *n* = 3 and 4
875	Elongation vibrations of the Pb-O bond in [PbO_6_] structural units
900–1100	Elongation vibrations of the Pb-O bond in [PbO_3_] structural units

**Table 3 materials-15-08061-t003:** Parameters (location of the center, C and integral intensity, A of IR peak) of deconvolution of the FTIR spectra of the MnO_2_-xLead vitroceramic materials, where x = 0–100 mol% Pb.

Samples		Positions and Integral Intensity of the IR Peaks					
Assignments of the Detected Peaks		Pb-O-Pb	[MnO_6_]	Mn-O-Mn	[PbO_n_]	[PbO_6_]	[PbO_3_]
x = 0%	C [cm^−1^] A [a.u.]	476 5.75	550 2.28	554 0.21	688 0.83	870 6.005	892 12.26
x = 10%	C [cm^−1^] A [a.u.]	475 6	520 3	581 3	680 2	840 10	923 15
x = 20%	C [cm^−1^] A [a.u.]	462 1.44	520 8	580 1	700 3	832 19.32	913 26.31
x = 30%	C [cm^−1^] A [a.u.]	470 6	520 4.1	688 4.5	752 2.22	870 12.01	950 16.03
x = 40%	C [cm^−1^] A [a.u.]	470.2 6.01	520 4.1	689 4.55	752.3 2.23	870 20.22	950 16.05
x = 50%	C [cm^−1^] A [a.u.]	470 7.11	553 4.56	600 1.01	778 5.15	870 25.55	970 2.02
x = 60%	C [cm^−1^] A [a.u.]	476 7.88	520 5.55	590 6.56	700 2.34	870 19.9	950 22.43
x = 70%	C [cm^−1^] A [a.u.]	475 8.88	520 4.55	600 1.54	680 10.66	875 16.78	970 23.45
x= 80%	C [cm^−1^] A [a.u.]	470 9.98	550 12.33	600 1.12	785 15.5	870 27.7	950 33.5
x = 90%	C [cm^−1^] A [a.u.]	486 7.78	520 14.89	600 2.3	720 1.12	877 28.9	950 40.5
x = 100%	C [cm^−1^] A [a.u.]	475 2.34	520 3.44	585 2.11	750 1.11	875 23.67	950 42.30

## Data Availability

Not applicable.

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
