# Peer review of "Structural, Optical, and Magnetic Studies of the Metallic Lead Effect on MnO2-Pb-PbO2 Vitroceramics"

_materials, 2022, doi:10.3390/ma15228061_

Round 1

Reviewer 1 Report

The proposed paper “Structural, optical and magnetic studies of the metallic lead effect on lead vitroceramics containing manganese dioxide”  tries to analyze the influence of glass composition on their structure.

Experimental section lacks crucial information about conducted studies – measurement technique (FT-IR), parameters e.g. resolution, scan speed, XRD step etc.

Presented structural data - XRD, FT-IR and UV-VIS indicate that regardless of the PbO2 / Pb ratio, the data looks exactly the same. This clearly suggests that this parameter has no influence on the structure of the tested materials, which is further confirmed by the energy gap measurements.

All of the FT-IR spectra should be deconvoluted into component bands and on this basis, the evolution of structure should be carried out.

Additionally, figure 2 (XRD) is completely illegible.

This type of data presentation is unacceptable.

Author Response

Reviewer 1

  1. The proposed paper “Structural, optical and magnetic studies of the metallic lead effect on lead vitroceramics containing manganese dioxide”  tries to analyze the influence of glass composition on their structure.Experimental section lacks crucial information about conducted studies – measurement technique (FT-IR), parameters e.g. resolution, scan speed, XRD step etc.

Authors

  1. The experimental section was improved in the revised manuscript.

Reviewer 1

  1. Presented structural data - XRD, FT-IR and UV-VIS indicate that regardless of the PbO2 / Pb ratio, the data looks exactly the same. This clearly suggests that this parameter has no influence on the structure of the tested materials, which is further confirmed by the energy gap measurements.All of the FT-IR spectra should be deconvoluted into component bands and on this basis, the evolution of structure should be carried out.

Authors

  1. The XRD, IR, UV-Vis, EPR data and the values of gap energies show that the PbO2/Pb ratio influence the structure of the vitroceramic.The deconvolution of the FTIR spectra was added in the revised manuscript.

Reviewer 1

  1. Additionally, figure 2 (XRD) is completely illegible.This type of data presentation is unacceptable.

Authors

  1. The XRD data were modified in the revised manuscript.

Reviewer 2 Report

The manuscript needs a complete revision of the English language

all the experimental parts should be written again with more obvious expressions to explain the results clearly and precisely 

Author Response

Reviewer 2

  1. The manuscript needs a complete revision of the English language.

Authors

  1. The English language was checked in the revised manuscript.

Reviewer 2

  1. All the experimental parts should be written again with more obvious expressions to explain the results clearly and precisely.

Authors

  1. The experimental parts were written detailed in the revised manuscript.

Reviewer 3 Report

In the present work, authors reported a detailed spectroscopic study of compositional variation of MnO2-xLead vitroceramic materials, and then investigated their structural, optical and magnetic properties. The results indicated that both optical and magnetic of samples varied with the addition of Pb content. A series of results and discussion were reasonable. However, the innovation in this paper is not very well put forward, some issues should be addressed.

1, The authors should elaborate the general applicability of the current work in Abstract section.

2, The introduction writing part need to be improved. Also, the writing and presentation of the introduction lacks a bit in clarity. The paper requires some amount of rewriting to clarify all aspects of it, especially the novelty and new findings of this work that need to be clearly mentioned. The authors have mentioned " In order to better understand manganese – lead glasses and vitroceramics from an application point of view, the study of structure is necessary...."…if this is the motivation of the current work, this point needs to be elaborated with existing research work aligned to this direction.

3, Authors may rearrange/polish the text and elaborated " Experimental Procedure " section the way so anybody can repeat the procedures, like a recipe. If there is process flow diagram can be added in fig. 1, it would be helpful to non-specialist readers (optional).

4, Controllable and rational processing is a determinant in magnetic properties of particles. How to adjust the magnetic properties of magnetic particles by modulating the processing in this work? The authors should also pay attention to this challenge, and some pioneering and original researches about controllable modulation of dielectric properties are suggested: Journal of Materials Chemistry C, 2016, 4, 9738; Composites Part A, 2018, 115, 371.

5, In XRD section, all the characteristic peaks should be indexed according to standard card. In addition, it is suggested to calculate the crystalline sizes of all samples, since the crystalline states and sizes have a great effect on the magnetic and optical properties.

6, I suggest authors conduct the morphological investigation of samples such as scanning electron microscope or others since the morphological characters show significant influence on the particle properties.

7, To what is due magnetic performance (magnetism, resonance or others...)? This fundamental issue is not all answered.

Author Response

Reviewer 3

1.In the present work, authors reported a detailed spectroscopic study of compositional variation of MnO2-xLead vitroceramic materials, and then investigated their structural, optical and magnetic properties. The results indicated that both optical and magnetic of samples varied with the addition of Pb content. A series of results and discussion were reasonable. However, the innovation in this paper is not very well put forward, some issues should be addressed. The authors should elaborate the general applicability of the current work in Abstract section.

Authors

  1. 1. The innovation and applicability of this paper were written in the abstract section.

Reviewer 3

  1. The introduction writing part need to be improved. Also, the writing and presentation of the introduction lacks a bit in clarity. The paper requires some amount of rewriting to clarify all aspects of it, especially the novelty and new findings of this work that need to be clearly mentioned. The authors have mentioned " In order to better understand manganese – lead glasses and vitroceramics from an application point of view, the study of structure is necessary...."…if this is the motivation of the current work, this point needs to be elaborated with existing research work aligned to this direction.

 Authors

  1. 2. The introduction section and the motivation of this paper were improved.

Reviewer 3

  1. 3. Authors may rearrange/polish the text and elaborated " Experimental Procedure " section the way so anybody can repeat the procedures, like a recipe. If there is process flow diagram can be added in fig. 1, it would be helpful to non-specialist readers (optional).

 Authors

  1. The flow diagram was added in the Figure 1. The experimental procedure was rewritten in the revised manuscript.

Reviewer 3

  1. Controllable and rational processing is a determinant in magnetic properties of particles. How to adjust the magnetic properties of magnetic particles by modulating the processing in this work? The authors should also pay attention to this challenge, and some pioneering and original researches about controllable modulation of dielectric properties are suggested: Journal of Materials Chemistry C, 2016, 4, 9738; Composites Part A, 2018, 115, 371.

  Authors

  1. 4. The controllable processes of the magnetic particles are not studied in this paper.

Reviewer 3

  1. In XRD section, all the characteristic peaks should be indexed according to standard card. In addition, it is suggested to calculate the crystalline sizes of all samples, since the crystalline states and sizes have a great effect on the magnetic and optical properties. I suggest authors conduct the morphological investigation of samples such as scanning electron microscope or others since the morphological characters show significant influence on the particle properties.

 Authors

  1. In XRD patterns the characteristic peaks were indexed according to standard card. The PDF number of the card was written in the revised manuscript. The sizes of crystallites are not subjected of the present paper. In the future we will investigate the size of crystallites and morphological characters (by scanning electron microscopy).

Reviewer 3

  1. To what is due magnetic performance (magnetism, resonance or others...)? This fundamental issue is not all answered.

 Authors

  1. The magnetic performance can be due to the paramagnetic behavior of the manganese ions.

Round 2

Reviewer 1 Report

FT-IR experimental misses one key information - which technique was used.

Where are the deconvolution parameters? What program was used?

Where are at least 2 deconvoluted spectra?

Author Response

Reviewer

  1. FT-IR experimental misses one key information - which technique was used. Where are the deconvolution parameters? What program was used? Where are at least 2 deconvoluted spectra?

Authors

  1. A deconvolution procedure of IR spectra was performed using Gaussian type function and Origin software. The parameters of deconvolution, namely location of the center, C and relative area, A of detected IR peaks of deconvolution with fit multi-peaks are summarized in Table 3.

Reviewer 2 Report

the manuscript can be published in journal of materials 

Author Response

The authors thank the reviewer for their extremely useful comments.

Reviewer 3 Report

Issues were addressed. 

Author Response

(The authors gave the same response as above.)

Round 3

Reviewer 1 Report

The authors did not present any deconvoluted spectra.

Presented FT-IR data is not sufficient enough to backup authors claims.

Furthermore, in spectroscopy, we don't use a term area, but integral intensity.

Presented parameters are not sufficient - what was RMS noise, what algorithm was used...

Author Response

Reviewer

The authors did not present any deconvoluted spectra.

Presented FT-IR data is not sufficient enough to backup authors claims.

Furthermore, in spectroscopy, we don't use a term area, but integral intensity.

Presented parameters are not sufficient - what was RMS noise, what algorithm was used...

Authors

The deconvoluted IR spectrum of the sample with x = 30 mol% Pb was shown in the Figure 3c in the revised manuscript.

In the Table 3 the area was substituted by integral intensity.

The R2 and Chi2 parameters were indicated in the Figure 3c. RMS noise did not make the subject of this manuscript.
